# Application of CO_2_-Soluble Polymer-Based Blowing Agent to Improve Supercritical CO_2_ Replacement in Low-Permeability Fractured Reservoirs

**DOI:** 10.3390/polym16152191

**Published:** 2024-08-01

**Authors:** Mingxi Liu, Kaoping Song, Longxin Wang, Hong Fu, Jiayi Zhu

**Affiliations:** 1Unconventional Petroleum Research Institute, China University of Petroleum, Beijing 102249, China; 1995liumingxi@sina.com (M.L.); 15717673116@163.com (L.W.); catherinefu90@sina.com (H.F.); 2College of Petroleum Engineering, Xi’an Shiyou University, Xi’an 710065, China; 18813109909@163.com

**Keywords:** fractured core, supercritical CO_2_, CO_2_-soluble blowing agent, static foaming performance evaluation, blocking rate, enhanced recovery, injection parameter optimization

## Abstract

Since reservoirs with permeability less than 10 mD are characterized by high injection difficulty, high-pressure drop loss, and low pore throat mobilization during the water drive process, CO_2_ is often used for development in actual production to reduce the injection difficulty and carbon emission simultaneously. However, microfractures are usually developed in low-permeability reservoirs, which further reduces the injection difficulty of the driving medium. At the same time, this makes the injected gas flow very fast, while the gas utilization rate is low, resulting in a low degree of recovery. This paper conducted a series of studies on the displacement effect of CO_2_-soluble foaming systems in low-permeability fractured reservoirs (the permeability of the core matrix is about 0.25 mD). For the two CO_2_-soluble blowing agents CG-1 and CG-2, the effects of the CO_2_ phase state, water content, and oil content on static foaming performance were first investigated; then, a more effective blowing agent was preferred for the replacement experiments according to the foaming results; and finally, the effects of the blowing agents on sealing and improving the recovery degree of a fully open fractured core were investigated at different injection rates and concentrations, and the injection parameters were optimized. The results show that CG-1 still has good foaming performance under low water volume and various oil contents and can be used in subsequent fractured core replacement experiments. After selecting the injection rate and concentration, the blowing agent can be used in subsequent fractured cores under injection conditions of 0.6 mL/min and 2.80%. In injection conditions, the foaming agent can achieve an 83.7% blocking rate and improve the extraction degree by 12.02%. The research content of this paper can provide data support for the application effect of a CO_2_-soluble blowing agent in a fractured core.

## 1. Introduction

With the exploitation of conventional oil reservoirs gradually entering the bottleneck period, many old oil fields have progressively entered the stage of high water content and low extraction degree. Unconventional reservoirs have gradually become the focus of petroleum exploration and development, in which the efficient exploitation of low-permeability reservoirs with permeability less than 50 mD is the current key research direction. In this type of reservoir, because of its small reservoir permeability, the water injection process pressure loss is large, and an imbalance of injection and production often occurs, so the conventional water drive development effect is poor. This type of reservoir is usually extracted by fracturing to produce artificial fractures. Although the existence of cracks exacerbates the flow of oil repellents, it also effectively reduces the differential pressure of the drive, dramatically reduces the energy loss, and helps transport the oil repellents to the deep part of the reservoir [1]. For reservoirs with a 10–50 mD permeability, combined with fracturing and reasonable well network deployment, conventional water drive can effectively utilize the crude oil in the reservoir [2,3]. However, for reservoirs with permeability less than 10 mD, the pore size of the core is too small. Under the combined influence of capillary force and oil–water interfacial tension, the water needs higher pressure to enter the interior of the pore throat, and the seepage resistance is higher. However, in a fracture–pore seepage system, water flows along the fracture surface where the seepage resistance is extremely low, and the proportion of water that can enter the interior of the pore throat for displacement is extremely low, making water drive utilization of fracture surface crude oil challenging. Compared with water, supercritical CO_2_ has extremely low viscosity and strong diffusivity, which can diffuse into core matrix pores and extract light components, so CO_2_ drive can effectively develop such reservoirs.

For reservoirs with less fracture development, fractures can be created by fracturing. The existence of artificial fractures dramatically improves the contact area between CO_2_ and rock. This facilitates the transport of CO_2_ to the deeper part of the reservoir, achieving better development results. The presence of fractures significantly increases the CO_2_ wave volume. CO_2_ expands the pore-throat utilization interval while significantly reducing the lower limit of pore-throat utilization. The presence of both simultaneously improves the degree of recovery in low-permeability reservoirs [4,5,6,7,8]. Currently, scholars believe that CO_2_ in fractures mainly diffuses in two modes of movement as follows: CO_2_ displaces crude oil in fractures while moving crude oil in the matrix through the swelling and extraction of crude oil [9]. As the permeability of the matrix increases, the diffusion phenomenon becomes more evident. Previous experiments found that CO_2_ can move part of the 30–100 nm tiny pore throats. If the proportion of <10 nm pore throats in some cores is higher, the diffusion effect of CO_2_ is seriously weakened, and it is more difficult to move [10]. Although CO_2_ has better dissolution and extraction properties, the fracture morphology directly determines the CO_2_ wave area, which directly affects the final extraction degree of the CO_2_ drive. In the actual development process, it is common to use the fracturing of the seam network to expand the contact area between CO_2_ and the fracture surface [11,12]. However, the existence of cracks will produce severe gas flushing, which will not only make the gas–oil ratio at the extraction end unpredictable but also make the reservoir pressure drop rapidly [13]; therefore, the gas flushing control of fractured reservoirs is also significant in actual production.

In order to solve the problem of gas flushing, scholars first proposed the alternating water–gas CO_2_ injection process. This method effectively maintains the reservoir pressure, facilitates the diffusion of CO_2_ into the matrix, and has a specific effect on controlling gas flushing in medium-high permeability reservoirs above 50 mD [14,15]. In order to control gas flushing in fractured reservoirs more effectively, scholars have successively developed blocking agents such as gels and microspheres [16,17]. For example, Yakai Li et al. developed a new type of high-temperature slow-release gel, which improved the recovery degree by 20% in a core with matrix permeability of 20 mD and fracture opening of 150 μm [18]. Haizhuang Jiang et al. used a new type of functional polymer to block a core with a matrix permeability of 20 mD and fracture permeability of 10,000 mD, which improved the recovery degree by 23.63% in the late stage of the CO_2_ drive [19]. Chengli Zhang et al. developed a salt-resistant polymer–surfactant synergistic composite low interfacial tension foam system, which increased the development degree by 13.17% after blocking the low-permeability reservoir of 10 mD [20]. Weiyao Zhu et al. developed a novel nano-microsphere system, which temporarily sealed the fractured core through the bridge and retention of the microsphere particles [21].

Moreover, retention temporarily seals the cracks and controls the oil repellent’s whole flow field, which regulates the gas flow in the subsequent gas drive process. With the advancement of technology, scholars have developed new blocking agents such as CO_2_ corresponding gel and CO_2_ corrosion-resistant microsphere particles, which are more targeted and have achieved good effects in blocking and improving the degree of recovery in CO_2_ drive experiments in fractured cores with permeability more significant than 1 mD [21,22]. For reservoirs with widely developed fractures at various scales, scholars injected gels, polymers, microsphere particles, and other agents sequentially to achieve the sequential blocking of fractures from large to small, and with the implementation of alternating water and gas injection and other implementation processes, they also achieved better blocking and improved recovery [8,23,24].

However, later studies found that in low-permeability fractured reservoirs with rock matrix permeability less than 1 mD, the presence of water in the fracture is detrimental to the degree of CO_2_ recovery, and the higher water saturation in the fracture reduces the ability of CO_2_ diffusion into the matrix; therefore, water acts as an inhibitor to CO_2_ drive [25,26]. Xu Li et al. developed a new water-soluble thixotropic gel system, which had good stability and good sealing performance in the matrix permeability of 0.96 mD and fracture openings of 15 mm fractured rock cores, but the gas drive did not improve the degree of extraction after sealing [27]. Qipeng Ma et al. measured the relative permeability curve of the nano-microsphere system in a low-permeability homogeneous core and experimentally found that in a core with a permeability greater than 10 mD, the dispersed system could effectively expand the two-phase oil–water flow. However, in cores with permeability less than 10 mD, the system reduced the water-phase permeability but did not improve the degree of recovery [28]. Daijun Du et al. used microsphere particles to seal fractured cores with a matrix permeability of 0.1–1 mD and a fracture permeability of 5000 mD; they also found that microspheres had an excellent blocking performance for a fracture after the experiments. However, the subsequent replacement was ineffective in increasing the degree of extraction [29]. This indicates that a fractured core with matrix permeability of less than 1 mD is not suitable for water drive development, and it is also not suitable for use in water-mediated agents for plugging.

In order to increase the degree of matrix development while fracture sealing in extra-low permeability fractured cores (matrix permeability less than 1 mD), some scholars have used a direct injection of CO_2_ foam for the replacement. From the macroscopic point of view, the presence of foam increases the injection pressure and promotes the diffusion ratio of the replacement medium from the fracture matrix while making the gas flushing rate decrease [30]; from the microscopic point of view, the injected foam forms resistance through the Jamin effect, which allows the replacement medium to drive off the crude oil within the pores of the fracture face [31]. In practice, factors such as the roughness of the fracture surface, injection flow rate, agent concentration, and total injected volume affect the sealing performance of the foam and must be evaluated comprehensively [32]. The stability of surfactant foaming alone is sometimes not very good, which reduces the blocking performance. Methods such as mixing CO_2_ and N_2_ foaming and adding nanoparticles to strengthen the surface properties of the foam have achieved good results in core replacement experiments [33,34,35,36].

Although the direct injection of foam can simultaneously reduce the rate of gas flushing and increase the degree of core recovery, the injectability of the system is relatively poor, resulting in higher injection pressure, higher energy loss, a smaller range of foam recovery, and other unfavorable phenomena [37]. In order to solve this problem, scholars developed a foaming agent that can be dissolved in both water and CO_2_, compared with the traditional water-soluble surfactant, gas–water amphiphilic surfactant dissolved in CO_2_ and injected into the core, and the foam formed with the water inside the fracture surface, significantly accelerating the expansion rate and area of the agent. The injection capacity is greatly improved [38] through continuous CO_2_ injection in the leading edge of the foam, which can be generated at the leading edge of the exfoliation by successive CO_2_ injections. Even if defoamed, foam can be generated again with water under subsequent CO_2_ injections [39]. The solubility of the agent in water and CO_2_ is different, and high solubility in water enhances adsorption during transport [40], so alcohols are often used as co-solvents to improve solubility in CO_2_ in experiments. Alcohols significantly increase the interaction between CO_2_ and polymer molecular chains, with ethanol having the most significant effect [41]. Indoor experiments showed that adding ethanol improved the foaming ability and stability of the agent. This helped foam regeneration [42], but it should be noted that the high solubility of the agent in CO_2_ can lead to poor foam stability, which is not conducive to blocking and mining. The amount of ethanol needs to be controlled [43]. Poor foam stability occurs when some blowing agents come into contact with crude oil [44]. The actual blocking performance and enhanced extraction are controlled by temperature, pressure, oil and water content, fracture morphology, agent concentration, and other factors, so the agent’s effectiveness must be evaluated through actual replacement experiments [45].

With the gradual decrease in global oil resources, extra-low permeability oil and gas reservoirs with K < 1 mD have attracted much attention in recent years. Because of the pore structure, oil–water interfacial tension, and other factors, such reservoirs cannot be developed by water drive, and supercritical CO_2_ has become the focus of current research because of its superior performance. To improve the wave volume of oil repellent in low-permeability reservoirs, supercritical CO_2_ fracturing is currently the mainstream reservoir modification method (hydraulic fracturing is not applicable). Therefore, controlling the gas flow in the fracture after fracturing and improving the recovery of CO_2_ drive is a problem that must be faced when developing such reservoirs. 

In this paper, the foaming performance of two kinds of CO_2_-soluble blowing agents in different CO_2_ phases, different water contents, and different oil contents is investigated for an extra-low permeability reservoir with K < 1 mD. Then, a more excellent blowing agent is screened according to the foaming situation, and a reasonable preparation method for this blowing agent is determined. Finally, the screened blowing agent is used to conduct driving experiments in a fractured core to study the effects of different injection speeds and injection concentrations on fractured core blocking and improving the degree of extraction. Good results were achieved, which have good prospects for practical application.

## 2. Materials and Methods

### 2.1. Experimental Material

Core and fracture pressing methods: Fracture morphology is one of the determining factors for the pattern of displacement, and in actual reservoirs, fracture openings, smoothness, and orientation can have a direct impact on the development effect. Since rocks with microfractures could not be drilled out of the completed core column, this paper’s indoor physical simulation tests used homogeneous, non-fractured, natural, low-permeability rocks to create seams after drilling and polishing the ends. Currently, the primary method to create fractures is by cutting the core or Brazilian splitting. Cutting the core directly minimizes the differences among different fractures and provides excellent controllability, but, at the same time, the surface of the fractures is exceptionally smooth, which is not conducive to foaming the system in the core. Brazilian splitting creates fractures by applying shear stress to the cores. Although it is difficult to ensure that the fractures among different cores are precisely the same since the cores come from the same stone, the rhythms and the degree of cementation are incredibly close to each other, the similarity in the fractures pressed out is exceptionally high, and the rough and uneven surface of the fractures is more conducive to foaming the agent in the formation, which is also in line with the situation in actual reservoirs. Therefore, Brazilian splitting was used to create fractures.

Fracture morphology: Since a full open fracture is best controlled during fracture with minimal variation among cores, all cores were pressed out of a full open fracture for experimentation in the later evaluation process (as shown in Figure 1).

Foaming agent, gas, experimental water, and experimental oil: Because of the low permeability of the matrix of the target core in this paper, which is about 0.25 mD, based on the results of previous studies, this type of core is not suitable for water injection development, and, likewise, it is not suitable for injecting water-soluble agents. Therefore, two high-temperature-resistant, high-salt CO_2_-soluble foaming agents were selected for this study, and the foaming method in the core was adopted for sealing. The gas was high-pressure CO_2_ with a purity of 99.9%, which was injected into the core after mixing the blowing agent and pressurizing to the specified pressure. In order to simulate the real stratigraphic situation and examine the actual blocking performance of the agent, the experiment was carried out using mineralized water with mineralization of 9800 mL/L for saturated cores; the specific formulations are shown in Table 1 below. The experiment was conducted with simulated oil, and the viscosity was 2.135 mPa·s under 25 °C/0.1 MPa. (The experiment was conducted for a low-permeability fractured reservoir in the Bohai Sea, with a matrix permeability of 0.1–0.5 mD and NaHCO_3_-type formation water, which needs to be developed twice after experiencing the pre-supercritical CO_2_ drive. This resulted in a decline in reservoir temperature and pressure, severe gas flushing, and a low extraction level).

The inorganic salts and white oil were purchased from Shanghai Aladdin Reagent Co. (Shanghai, China). All equipment from Yongruida Technology Co. (Beijing, China). All CO_2_ comes from Shunxing Co. (Beijing, China).

### 2.2. Experimental Methods

#### 2.2.1. Evaluation Methods and Test Programs for the Foaming Capacity of Blowing Agents

Experimental program: The foaming performance with liquid CO_2_ and supercritical CO_2_ was first investigated for the two blowing agents. The liquid CO_2_ foaming temperature and pressure were 22 °C and 10 MPa; the supercritical CO_2_ foaming temperature and pressure were 70 °C and 20 MPa, and the water content was 10% in all cases (the volume of the foaming instrument was 250 mL, and the water volume was 25 mL). The concentration of the agent was 1.6% (CO_2_ and water were solvents in the system, the volume of solvent was 250 mL, the foaming agent was 4 mL, and the concentration was 1.6%). The water content in the supercritical CO_2_ foaming system was subsequently varied, with the blowing agent content remaining constant at 1.6%, and the foaming performance was investigated for water contents of 5%, 10%, and 20% (12.5, 25, and 50 mL of water, respectively). Finally, water–oil ratios of 100:3, 100:10, and 100:20 were set at 10% water content to investigate the effect of oil content on the foaming performance of the supercritical CO_2_ foaming system and to conduct foaming agent screening.

A visual high-temperature and high-pressure foaming apparatus was used to evaluate the foaming performance of the agents. The experimental steps and experimental flowchart are shown in Figure 2 and described as follows:
First, the solution was configured in the foaming apparatus according to the concentration of the agent, water content, and oil content required by the experiment. Then, the foaming apparatus was closed and sealed, followed by vacuuming for 15 min (to avoid the loss of the foaming agent and oil, the solution was prepared directly inside the instrumentation; to avoid the influence of air, the equipment was evacuated prior to CO_2_ injection);Since the gas pressure will rise sharply during the heating process, to ensure the experiment’s safety, do not increase the pressure before the end of the heating. If the pressure is not reached at the end of the heating process, the pressure should be increased slowly and not too quickly.After the temperature and pressure in the equipment were stabilized, the stirring switch was turned on. After the foaming was stabilized, the foaming performance was examined according to the foaming height and half-life of the agent, and the agent screening was completed.

#### 2.2.2. Methods and Experimental Protocols for CO_2_ + Blowing Agent Exfoliation of Fractured Cores

Experimental protocol: Prior to the sealing experiments, pure CO_2_ replacement experiments (as a control test) were first carried out on fractured cores at different injection flow rates (injection rates of 0.1, 0.2, 0.4, 0.6, and 0.8 mL/min), followed by CO_2_ + foaming agent replacement experiments. In order to investigate the sealing of the fractured core by the blowing agent after CO_2_ replacement and the ability to increase the degree of extraction, the experiment was divided into three segment plugs with CO_2_-CO_2_ + blowing agent-CO_2_ injection, where CO_2_ was mixed with the blowing agent through a stirring piston. For the core with one fully open fracture, firstly, the driving effect of the blowing agent at different injection rates was investigated for the preferred injection rate (injection concentration of 1.11%, injection rate of 0.1, 0.2, 0.4, 0.6, and 0.8 mL/min). The sealing of the fractured core and the enhanced recovery characteristics of the different concentrations of agents were then examined at preferred injection rates (blowing agent concentrations of 0.56%, 1.11%, 1.67%, 2.28%, and 2.80%, respectively). The experimental temperature was 70 °C, controlled by a thermostat; the experimental minimum pressure was 20 MPa, controlled by a pressure-return valve; and the experimental core was 3.8 cm in diameter and 8 cm in length.

The foaming agent configuration method and experimental steps are shown in Figure 3a, and the foam formation schematic is shown in Figure 3b. The steps are described as follows:The core was polished to the specified size, dried, and weighed, and then vacuumed and saturated with water. Because of the low permeability of the core, the vacuuming time needed to be more than 8 h, and the saturated formation water time needed to be more than 6 h. After the saturated water, the core was weighed, and the pore volume was calculated;Because of the large cross-sectional area of the core and low permeability, saturated oil was injected with 20 MPa constant pressure, and after about 24 h, the saturated oil was finished. At this time, the oil saturation degree of the core was about 50% (it was difficult to saturate low permeability cores with oil, and the saturation was only 55% after 72 h of saturation);The core is removed and immediately fractured at the end of saturated oil. A reasonable fracturing method is developed based on the fracture morphology, and the core is loaded into the gripper immediately after fracturing in preparation for deplacement (the fracturing process takes about 120 s, which is extremely short, and the pore sizes are small, the volatilisation of fluids in the core is very small, and the loss of oil and water during the fracturing process is negligible);After loading the core, the inlet and outlet of the gripper were closed, and 2 MPa perimeter pressure was applied, followed by heating. To ensure that the core reached the experimental temperature, it was heated for more than 4 h. To avoid the volatilization of oil and water from the side of the core during the heating period, it was necessary to add a perimeter pressure so that the sleeve stuck close to the surface of the core. At the same time, the outlet and the inlet of the gripper were closed to prevent the oil and water vapours that evaporated from the core’s end face from escaping out of the gripper;After heating, a hand pump was used to pressurize the pressure return valve to 20 MPa. The gas was then pressurized using a booster pump, and when the gas in the piston was 20 MPa, the pressurization was complete. Inside the thermostat, the gas was injected into the core through a metal coil of about 6 m in length. As the gas flowed through the high-temperature coil, it was sufficient to reach the specified temperature and complete the phase transition;After opening the inlet end of the gripper, the gas flow reached the core outlet immediately through the fracture, and the pressures at both ends were balanced after about 90 s. Subsequently, the injection flow rate was set, and the outlet end of the gripper was opened in preparation for the replacement (during the replacement process, the oil and gas production at any time during the stage time and the pressure data were recorded);The injection was stopped after injecting about 1–4 PV. At this time, the pure CO_2_ replacement experiment was finished; the thermostat was closed, and the pressure was removed. To switch to the CO_2_ + blowing agent, we closed the gripper inlet and outlet after stopping the injection and recorded the pressure inside the core;Current CO_2_-soluble agents use water as an intermediate medium, and CO_2_ can scramble the blowing agent out of the water during constant stirring. The stirred piston was therefore fitted with a propeller on one side and configured by pouring the aqueous solution directly into the piston, evacuating the air inside the piston using a vacuum pump (15 min), followed by injection and pressurization with CO_2_;After pressurizing to the specified pressure, the piston position was adjusted so that the propeller was in direct contact with the aqueous solution. The propeller was then opened and stirred for 20 min and left to stand for 20 min after the end of stirring. Subsequently, the core entrance and exit were opened, the flow rate was set after the pressure and gas flow rate stabilized, the CO_2_ + blowing agent drive was started, and the experimental data were recorded. At this time, the piston pressure was slightly higher than the internal pressure of the core, and there was a small pressure disturbance after opening the outlet, which stabilized after about 30 s.Step (9) was repeated after the end of the CO_2_ + blowing agent replacement. After stabilization, the subsequent pure CO_2_ replacement was turned on, and experimental data such as differential pressure, oil production, etc., were recorded. The experiment was completed after a certain amount of replacement.

## 3. Results and Discussion

### 3.1. Evaluation of the Foaming Capacity of CO_2_-Blowing Agents

Using the foaming equipment, two CO_2_-soluble blowing agents, CG-1 and CG-2, were selected to conduct experiments to evaluate the foaming ability under different CO_2_ phases and different water and oil contents.

In the process of stirring the aqueous solution with CO_2_, it does not foam immediately. In the case of the CG-1 foaming process with liquid CO_2_, for example, as shown in Figure 4, at the beginning of the mixing process, foam of a large size is formed, visible to the naked eye, and spreads throughout the entire viewport during the mixing process. The foam in the middle of stirring gradually sinks to the bottom of the instrument; the upper foam is large and the lower foam is small. In late stirring, all the foam sinks to the bottom of the container, forming a stable, fine foam, and only a tiny amount of foam exists hanging on the viewport. When the foaming was over, the stirrer was closed, and the foaming height was measured. After defoaming, the half-life was measured (the stirring process of the two blowing agents was about 90–120 s).

#### 3.1.1. Comparison of Foaming Ability of Agents in Different Phases

Liquid CO_2_ (22 °C/10 MPa): Figure 5, shows CG-1 in 10% water at a concentration of 1.6% and a foaming height of 1 cm. The foam volume is small but has excellent stability with a half-life of 450 min. For CG-2 in 10% water, a concentration of 1.6% is required to form a stable foam with a half-life of up to 1200 min, but the bubble height is still low at 2.7 cm.

Supercritical CO_2_ (70 °C, 20 MPa): Figure 6 shows CG-1 at 10% water and a concentration of 1.6% with a foaming height of 4.5 cm and a half-life of 100 min. CG-2 is shown at 10% water and a concentration of 1.6% with a foaming height of 5.3 cm and a half-life of 15 min.

Under the condition of low water quantity, both agents formed stable foam with liquid CO_2_ and supercritical CO_2_, but the foam state varied greatly. The phenomenon of lower foaming height and a longer half-life is commonly observed in liquid CO_2_, and in supercritical CO_2_, the foaming height is higher, but the half-life is significantly shorter. Because of the higher density of liquid CO_2_, the phenomenon of molecular thermal movement is weaker, and the volume of foam formed with water is significantly smaller, which is more stable under the maintenance of high pressure. Supercritical CO_2_ is less dense. The phenomenon of molecular thermal movement is weaker, which leads to a higher foaming height, and at the same time, because of the presence of high temperature, the stability of the foam is significantly weakened and the half-life decreases dramatically. Through the experiments, it was found that the foaming effect of CG-1 with supercritical CO_2_ was better, and the foaming effect of CG-2 with liquid CO_2_ was better.

According to the experimental results, the foaming volume of the blowing agent and liquid CO_2_ is low. Therefore, after mixing thoroughly by stirring, the foam occupies only a tiny part of the volume in the piston. The upper part of the foam is a mixture of CO_2_ and the blowing agent, and the foam will not be injected into the rock center during the replacement, which ensures a pure gas injection environment (as shown in Figure 7).

#### 3.1.2. Comparison of the Foaming Capacity of the Blowing Agent and Supercritical CO_2_ at Different Water Contents

Since supercritical CO_2_ was used to foam with water in the replacement experiments, focusing on the foaming performance of the agent with supercritical CO_2_ was necessary. The temperature and pressure of foaming remained unchanged at 70 °C and 20 MPa, and the concentration was 1.6%. Based on 10% water content (25 mL), two test points of 5% (12.5 mL) and 20% (50 mL) were added to examine the foaming performance of each agent.

Table 2 presents the foaming data of agents and supercritical CO_2_ under different water content conditions. It can be seen that the rise in water content plays a prominent role in promoting the foaming effect. Under the condition of 5% water content, the foaming height and half-life of CG-1 were shortened (0.4 cm/50 min), and CG-2 could not even foam. At 20% water content, the foaming height and half-life of CG-1 were 5.5 cm and 250 min, respectively, with a half-life 1.5 times that of 10% water, and the foaming height and half-life of CG-2 were 4.6 cm and 70 min, respectively, with a half-life 4.67 times that of 10% water. As shown in the foaming data, CG-1 also had a specific foaming ability in the case of low water content, and the scope of application was more comprehensive; CG-2 only had a better foaming effect in high water content conditions.

#### 3.1.3. Comparison of the Foaming Capacity of the Blowing Agent and Supercritical CO_2_ at Different Water Contents

Oil and water coexist in the rock pore throat, so the foaming effect of the agent in the case of oil content must be examined. Water–oil ratios of 100:3 (0.75 mL), 100:10 (2.5 mL), and 100:20 (5 mL) were set to investigate the effect of oil content on the foaming performance of the supercritical CO_2_ foaming system. The temperature and pressure were also 70 °C and 20 MPa, the concentration was 1.6%, and the water content was 10% (25 mL) unchanged. In order to measure the effect of oil content on foaming performance more accurately, the “foam factor” was introduced as a non-factorized quantity, where the higher the foam factor, the better the foaming performance (foam factor = 0.75 × foaming height × half-life).

As shown in Table 3, with the increase in oil content, the foam factor showed a tendency to increase and then decrease. In the case of low oil content, the oil on the two foaming agents showed a prominent promotion of foaming. The foaming effect of CG-2 was pronounced after adding oil, where the foam factor became 6–10 times the case of no oil, and the foaming performance increased significantly. The foam factor of CG-1 after refueling became 0.75–2.15 times the original, and there was a phenomenon of weakening the foaming effect at high oil content. The table shows that although the effect of oil on CG-1 is relatively tiny, overall, the foaming effect of CG-1 after refueling is still better than that of CG-2.

#### 3.1.4. Foaming Agent Screening

According to the results of the above studies, it can be found that the foaming effect of CG-2 with liquid CO_2_ is better than that of CG-1, but the foaming effect of CG-1 with supercritical CO_2_ is better than that of CG-2. CG-1 can be foamed under low water content conditions, and its foaming effect is better than that of CG-2 in various water content conditions. Although the promotion effect of oil on the foaming performance of CG-2 is better than that of CG-1, the oil content of CG-1 is better than that of CG-2. Although the promotion effect of oil on the foaming performance of CG-2 is better than that of CG-1, the foaming effect of CG-1 in oil content is better than that of CG-2 on the whole. Based on the above experimental results, it is assumed that CG-1 has a better foaming effect with supercritical CO_2_ and performs better under low water conditions. At the same time, CG-1 still maintains an excellent foaming effect under oil-containing conditions. Therefore, CG-1 was selected to investigate the effect of the use of the agent in fractured rock cores.

### 3.2. Effectiveness of CO_2_-Soluble Blowing Agent in Fractured Rock Cores

Because of the low permeability of the core matrix (about 0.25 mD), it is not possible to use water drive development. Therefore, the CO_2_-soluble foaming agent was dissolved in CO_2_ and injected into the core, which foamed with the oil and water on the fracture surface to form foam and then sealed the core, thus increasing the recovery. Cores with one fully open fracture were used for this study.

#### 3.2.1. Influence of the Injection Rate on the Repellent Characteristics of a Fully Open Fracture Core (Blank Controlled Experiment)

The gas flow rate graphs of the core with one fully open fracture at different injection rates are shown in Figure 8. With the increase in the injection flow rate from 0.05 to 0.8 mL/min, the gas production rate in the late stage of replacement gradually increased from 421.36 to 1326.86 mL/(MPa·min).

The effect of different injection flow rates on the degree of replacement from a fully open fracture core is shown in Figure 9. The figure shows that as the injection flow rate rises, the replacement degree shows a tendency to first increase and then decrease. In the replacement process of a fractured core, oil is first transported from the matrix into the fracture and subsequently carried out of the core by the CO_2_ + blowing agent. Too low a flow rate results in a low amount of CO_2_ being injected, while at the same time, the differential pressure is low (as shown in Figure 10, the higher the flow rate, the higher the differential pressure). The amount of diffusion into the fracture will be even lower, preventing the effective movement of crude oil from the matrix. However, if the flow rate is too high, although the differential pressure rises, in that case, the flow rate of the gas is too fast, which is also unfavorable to the diffusion of CO_2_ into the matrix, leading to a decrease in the degree of recovery.

#### 3.2.2. Effect of Injection Velocity on Displacement Characteristics of the Foaming Agent

Figure 11 shows the stage gas production rates at different injection rates for CO_2_ replacement of a core with one fully open fracture with both CG-1 concentrations of 1.11%. The graphs show the gas production rates when the injection rate increases from 0.1 mL/min to 0.8 mL/min. For the pure CO_2_ drive stage, the injection rate was 0.4 mL/min in all cases, and the stage gas production rate range was roughly 850–1200 mL/(MPa-min) because of the difference in fracture morphology. The gas production rate was suppressed immediately after injection of the blowing agent (including the experimental group of 0.4–0.8 mL/min), showing a gradual decrease in the gas production rate, which quickly returned to the previous state after reinjection of CO_2_.

Since the gas production rate is affected by both the injection rate of the system and the agent, the injection rate of the CO_2_ + foaming agent is not the same as that of pure CO_2_. Therefore, the resistance factor was used to examine the blocking performance of the agent by calculating the ratio of the injection pressure difference at the late stage of the replacement at the same injection rate (the pressure was unstable in the early stage of the replacement, and it took time for the foam to form). The resistance factor for foam formation at each injection rate is shown in Table 4.

Table 4 and Figure 11b show that the resistance factor increases and then decreases as the injection rate increases. The resistance factor was higher at 0.4 and 0.6 mL/min measuring 2.20 and 2.95, respectively. In the static foaming evaluation, foaming was carried out by stirring, which stopped after the foaming was completed. However, in the replacement process, without any stirring device, it can only rely on the flow of gas to provide power, in turn forming foam on the fracture surface, which is both the foam’s storage space and the gas’s main flow channel. Therefore, the increased gas flow both promotes foam formation and leads to foam destruction, so a decreasing drag factor at a high flow rate occurs.

#### 3.2.3. Influence of the Injection Concentration on the Effect of Expulsion

According to the results in Table 4, the injection flow rates of the agents, 0.4 mL/min and 0.6 mL/min, were preferably selected. The blowing agent concentrations were set to 0.56%, 1.11%, 1.67%, 2.28%, and 2.80% to investigate the influence of the blowing agent concentration on the effect of the replacement (0.4 mL/min in all cases of pure CO_2_ drive). Figure 12 shows a comparison of the stage gas production rate, resistance factor, and sealing rate at different injection concentrations when the agent injection rate is 0.4 mL/min (at this time, the injection rate of the CO_2_ + blowing agent is the same as that of the pure CO_2_ replacement, and the existence of the blocking rate is meaningful). It was found that at an injection rate of 0.4 mL/min, CG-1 had a superior sealing performance at all concentrations, and the sealing performance gradually increased with the increase in injection concentration. At the injection concentration of 2.8%, the resistance factor in the late replacement stage was 6.1, the blocking rate was 80.72%, and the stage gas production rate was 274 mL/(MPa·min).

Figure 13 shows a comparison of the stage gas production rate, resistance factor, and sealing rate at different injection concentrations at the agent injection rate of 0.6 mL/min. At an injection concentration of 2.8%, the resistance factor in the late replacement stage was 6.3, the sealing rate was 83.775%, and the stage gas production rate was 200 mL/(MPa-min). After increasing the injection rate from 0.4 mL/min to 0.6 mL/min, there was an increase in the sealing performance at all concentrations, and in terms of the sealing rate, the increase in the sealing effect by increasing the injection rate was not pronounced. However, from the early injection stage of the system, the onset time of the agent was shortened, the sealing was more rapid, the rate of gas production in the stage decreased more rapidly, and the fluctuation in the resistance factor was significantly weakened. It can be considered that the foam is more stable at 0.6 mL/min.

A comparison of CG-1 sealing rate at different injection rates and concentrations is shown in Figure 14a. At a CG-1 concentration of 0.56%, the injection flow rate of 0.4 mL/min has a better sealing effect; when the CG-1 concentration is more significant than 1.11%, it has a better sealing effect at an injection flow rate of 0.6 mL/min. Through static experiments, it was found that when the concentration is low, the foaming volume and stability are reduced. The injection rate is high, which promotes foaming, but it currently has a stronger destructive ability, leading to a decrease in the sealing rate, so it is suitable to use a low injection flow rate. The foam is stable at high concentrations and can resist higher injection flow rates, while a high flow rate promotes system foaming and strengthens the sealing performance.

A comparison of the degree of enhanced recovery of CG-1 at different injection rates and concentrations is shown in Figure 14b. Although a fully fractured core is relatively controllable during the fracturing process, ensuring complete consistency between cores is challenging, and the degree of recovery varies significantly among different cores. Therefore, the sealing performance was evaluated using the recovery enhancement after the agent was injected into the core. The enhancement of recovery of CG-1 was 1–3% when the concentration was less than 1.67%, but the enhancement of recovery was 6–12.5% when the concentration ranged from 1.67% to 2.80%, which had a significant effect. The sealing performance of CG-1 at a high injection concentration and a high flow rate was excellent, with a sealing rate above 80%, which significantly increased the proportion of CO_2_ diffusion into the matrix and, therefore, improved the degree of recovery significantly. In the case of high flow rate injection, the amount of CO_2_ injected into the core for the same length of time was significantly higher, increasing the injection differential pressure and, at the same time, increasing the proportion of CO_2_ diffusing into the core matrix. This led to a significant increase in the degree of increased recovery at the higher flow rate for two injection flow rates that are close to each other in terms of sealing rate.

## 4. Conclusions

This paper focuses on the foaming performance of CO_2_-soluble blowing agents under different conditions and their effectiveness in sealing the fractured core and enhancing recovery. Based on a comparison of the foaming effect of foaming agents in different phases, water contents, and oil contents, CG-1 is preferred for evaluating the replacement effect, which has a good use effect. The specific conclusions are as follows:Through static experimental evaluation, it is found that CG-1 can produce stable foam under low water content and various oil content conditions. The foaming conditions are more relaxed, with a broader range of use, which is suitable for use in reservoirs with variable conditions and can be applied to the subsequent evaluation of the replacement experiments.Because of the low foaming height of both blowing agents with liquid CO_2_, a large amount of CO_2_ and blowing agent mixture exists in the upper part of the space, which ensures that there is no participation of any foam or water in the injection process and provides a method for the configuration and injection of the agent.In the dynamic replacement experiment, if the injection rate is too low, the foaming agent cannot form stable foam with the water and oil on the fracture surface. However, if the injection rate is too high, the airflow will simultaneously destroy the foam that has already been formed, so that the sealing rate will be reduced, which is unfavorable to development. At 0.6 mL/min and 2.80% injection, the sealing rate can reach 83.7%, and the improvement in the extraction degree is 12.02%, which has a good use effect.

## Figures and Tables

**Figure 1 polymers-16-02191-f001:**
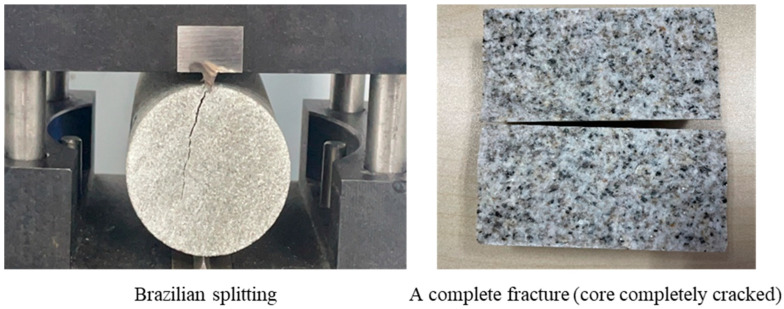
A Brazilian split and a fully open fracture core.

**Figure 2 polymers-16-02191-f002:**
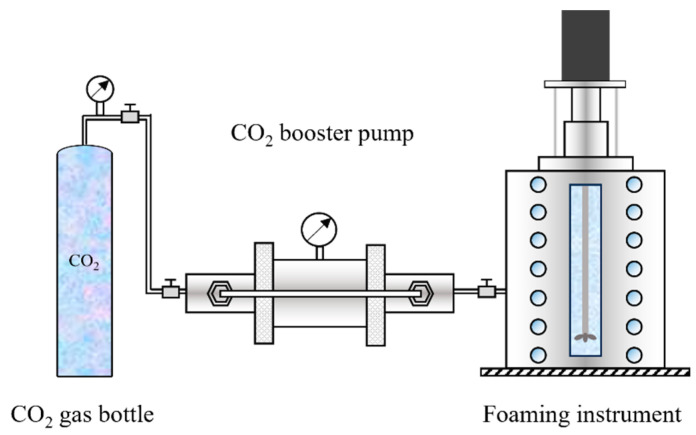
Flowchart of static evaluation experiment of foaming agent.

**Figure 3 polymers-16-02191-f003:**
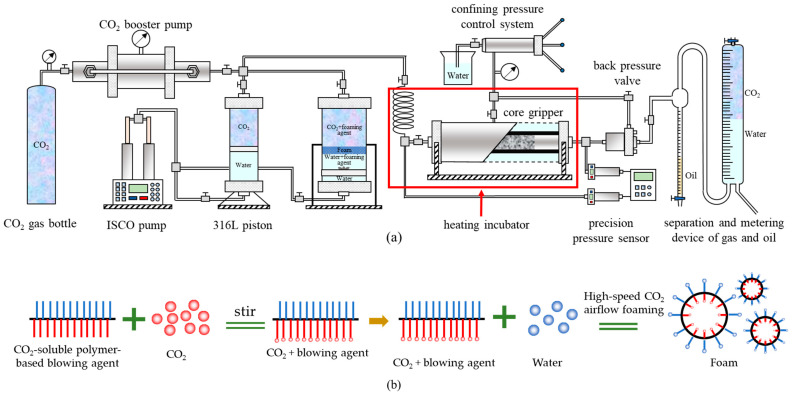
Flowchart of CO_2_ + blowing agent drive experiments on fractured cores (**a**) and schematic diagram of foam formation (**b**).

**Figure 4 polymers-16-02191-f004:**
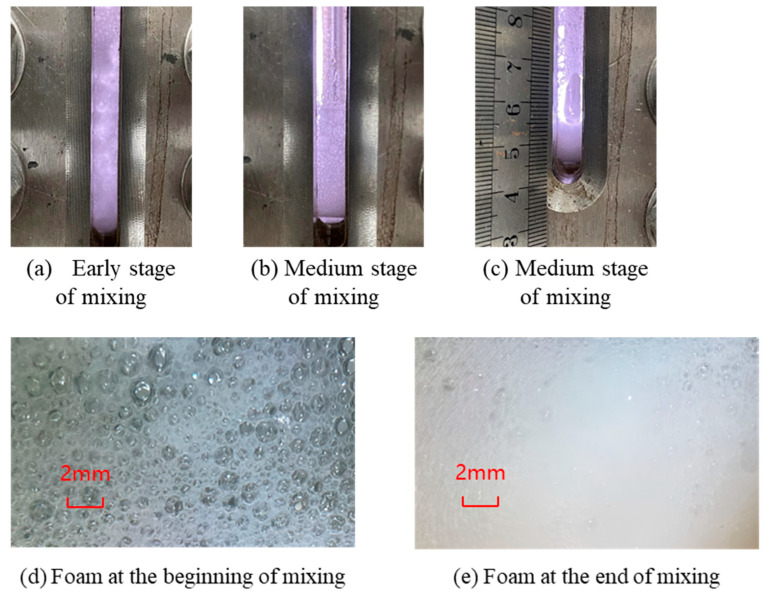
CG-1 and liquid CO_2_ foaming process.

**Figure 5 polymers-16-02191-f005:**
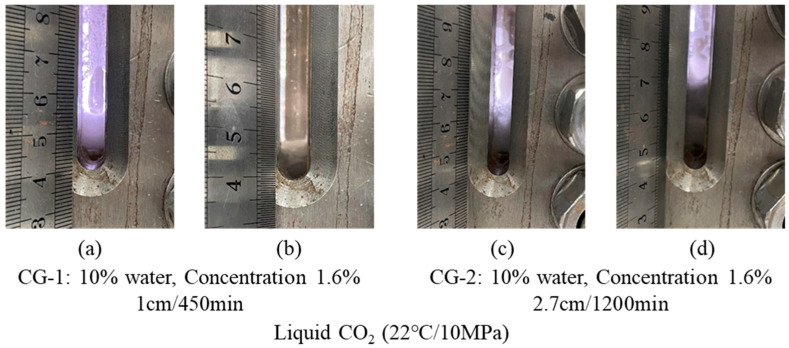
Results of CG-1 and CG-2 foaming with liquid CO_2._ (**a**) Photos of CG-1 at the end of foaming; (**b**) Photographs taken when CG-1 has reached its half-life; (**c**) Photos of CG-2 at the end of foaming; (**d**) Photographs of CG-2 when it has reached its half-life.

**Figure 6 polymers-16-02191-f006:**
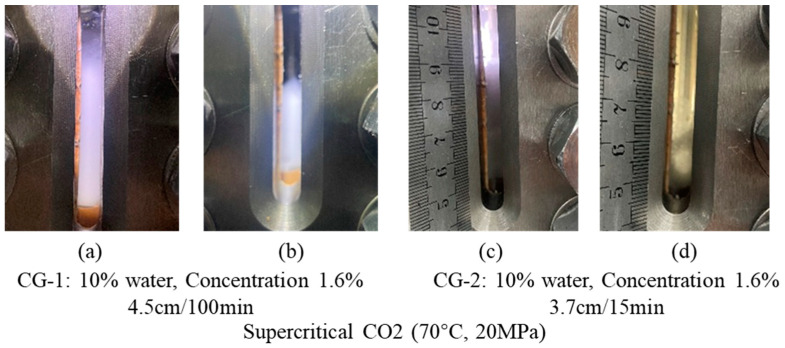
Results of CG-1 and CG-2 foaming with supercritical CO_2._ (**a**) Photos of CG-1 at the end of foaming; (**b**) Photographs taken when CG-1 has reached its half-life; (**c**) Photos of CG-2 at the end of foaming; (**d**) Photographs of CG-2 when it has reached its half-life.

**Figure 7 polymers-16-02191-f007:**
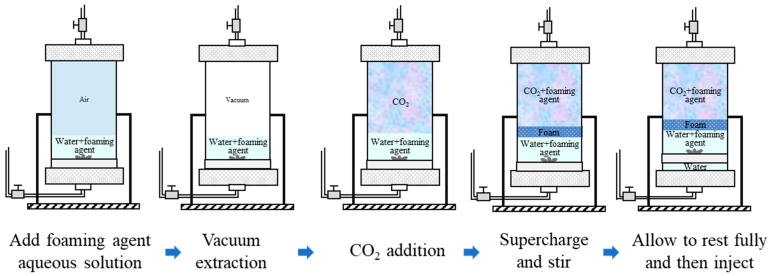
Configuration of CO_2_ + blowing agent mixtures.

**Figure 8 polymers-16-02191-f008:**
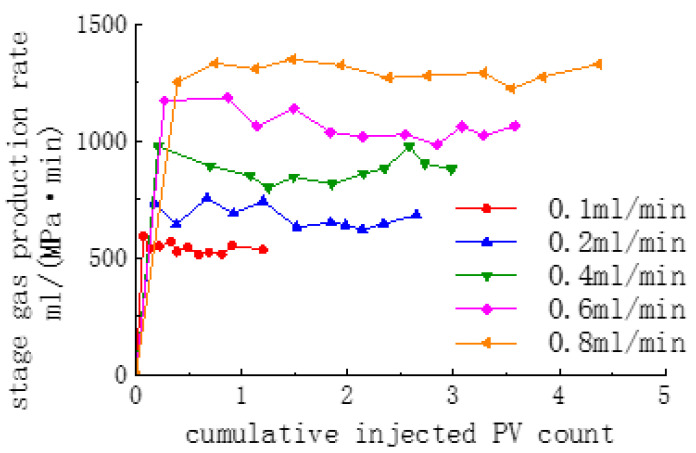
Gas flow rates at different injection rates.

**Figure 9 polymers-16-02191-f009:**
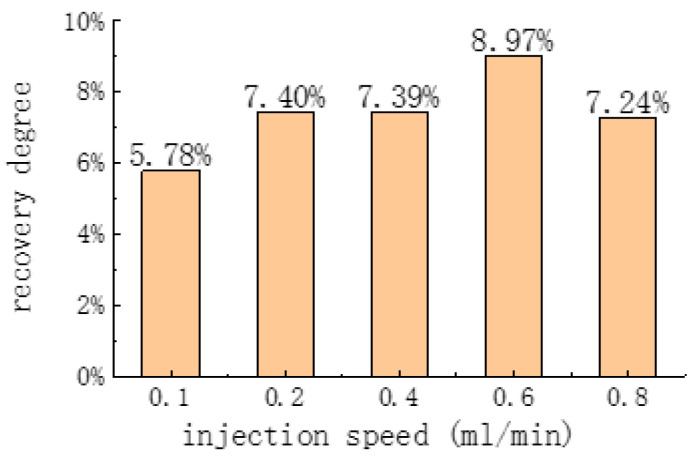
Influence of different injection velocities on the fracture core recovery degree.

**Figure 10 polymers-16-02191-f010:**
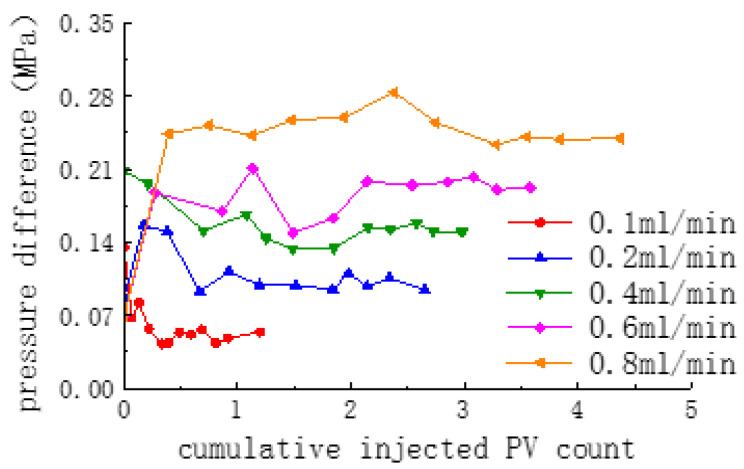
Displacement pressure difference at different injection flow rates.

**Figure 11 polymers-16-02191-f011:**
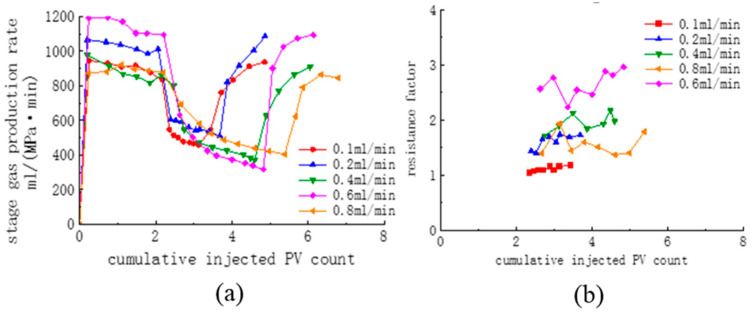
Stage gas production rates (**a**) and the resistance factor (**b**) at different injection rates.

**Figure 12 polymers-16-02191-f012:**
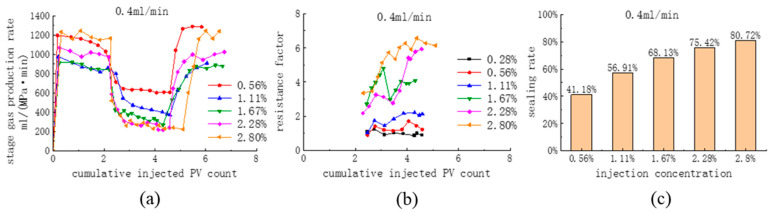
Comparison of (**a**) the stage gas production rate, (**b**) the drag factor, and (**c**) the blocking rate at different injection concentrations (0.4 mL/min).

**Figure 13 polymers-16-02191-f013:**
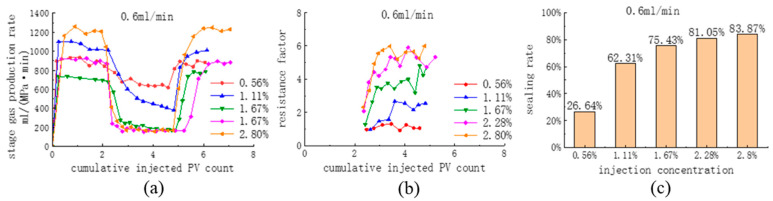
Comparison of (**a**) the stage gas production rate, (**b**) the drag factor, and (**c**) the blocking rate at different injection concentrations (0.6 mL/min).

**Figure 14 polymers-16-02191-f014:**
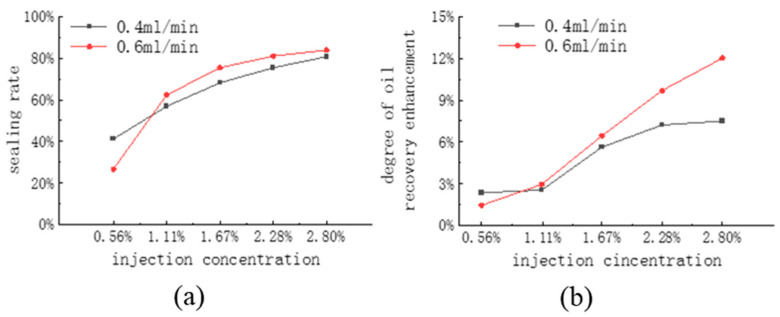
Comparison of CG-1 (**a**) the sealing rate and (**b**) enhanced recovery at different injection rates and concentrations.

**Table 1 polymers-16-02191-t001:** Ionic species and concentration of experimental water.

Ionic Species	K^+^ & Na^+^	Mg^2+^	Ca^2+^	Cl^−^	SO_4_^2−^	HCO_3_^−^	CO_3_^2−^
Ionic concentration mg/L	2704.30	11.87	68.49	880.50	20.45	4690.94	68.49
Total mineralization	8912.08 mg/L						

**Table 2 polymers-16-02191-t002:** Foaming effect of pharmaceuticals with supercritical CO_2_ under different water content conditions.

Sample Number	CO_2_ Phase	5% Water Content	10% Water Content	20% Water Content
CG-1	Supercritical	0.4 cm/50 min	4.5 cm/100 min	5.5 cm/250 min
CG-2	Supercritical	Non-foamable	3.7 cm/15 min	4.6 cm/70 min

**Table 3 polymers-16-02191-t003:** Foaming effect of pharmaceuticals with supercritical CO_2_ under different oil content conditions.

Sample Number	100:0	100:3	100:10	100:20
CG-1	4.5 cm/100 min 337.5	4.5 cm/120 min 405	4.8 cm/200 min 720	4.5 cm/75 min 253.125
CG-2	3.7 cm/15 min 41.625	4 cm/90 min 270	4.3 cm/140 min 451.5	4.6 cm/100 min 345

**Table 4 polymers-16-02191-t004:** Resistance factor for blowing agent formation at each injection rate.

injection rate mL/min	0.1	0.2	0.4	0.6	0.8
resistance factor	1.17	1.73	2.20	2.95	1.80

## Data Availability

The data presented in this study are available on request from the first author. (The data is not publicly available due to confidentiality policies).

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
