# Peer review of "Application of CO2-Soluble Polymer-Based Blowing Agent to Improve Supercritical CO2 Replacement in Low-Permeability Fractured Reservoirs"

_polymers, 2024, doi:10.3390/polym16152191_

Round 1

Reviewer 1 Report

Comments and Suggestions for Authors

1. In the Introduction, there are some deficiencies in the expression. Specifically, on Lines 50-52, the authors state that “However, for reservoirs with permeability less than 10mD, utilizing the crude oil on the fracture surface by water drive is challenging, and CO2 drive is often used for development.” This statement lacks an explanation of why CO2 drive is more suitable when water drive is challenging in reservoirs with low permeability. For instance, CO2can dissolve components in crude oil within subsurface reservoirs, thereby reducing the viscosity of the oil and enhancing its mobility, which facilitates oil displacement. Please supplement this statement by providing explanations to make it more complete and logically coherent. The same issue to the statement “For low-permeability reservoirs with less fracture development, CO2 fracturing is often used for development” on Lines 53-54.

2.  I recommend adding a paragraph in the Introduction section to provide a deeper discussion of the significance of this study. Emphasize that this research not only addresses the challenges in developing low-permeability reservoirs but also presents a practical CO2 utilization technology with promising real-world applications. Moreover, highlight how this study aligns with ongoing research in CCUS in China and globally, especially since 2023, thereby underscoring its broader relevance and importance.

3. Please review the entire manuscript and correct the subscripts for CO2 and N2.

4. In lines 197-200, the authors mentioned the content of mineralized water and listed the ionic species of the water in Table 1. Please provide additional information regarding the source of the water used in this study. Does it come from a specific reservoir? Please include details about the characteristics of this reservoir, including its rock properties, development history, and any distinctive features. This addition will help readers fully understand similar reservoirs where the results of this study might be applicable, facilitating the broader application and dissemination of the study’s related technologies and findings.

5.  Some sentences are quite long and complex, significantly affecting readability. For instance, on Lines 277-282, the sentence “in order to avoid the volatilization of oil and water from the side of the core during the heating period, it is necessary to add a perimeter pressure so that the sleeve is stick close to the surface of the core, and at the same time, close the outlet and the inlet of the gripper, to prevent the oil and water vapors that have been evaporated from the end face from escaping out of the gripper” contains 76 words. Breaking it into shorter, more manageable parts can enhance clarity and readability. The authors may consider revising it as follows: “To avoid the volatilization of oil and water from the sides of the core during the heating period, it is necessary to add perimeter pressure. This pressure ensures that the sleeve sticks closely to the surface of the core. Additionally, the outlet and the inlet of the gripper should be closed to prevent the evaporated oil and water vapors from escaping from the end face of the gripper.” Similar improvements can be applied to other lengthy sentences, such as those found on Lines 283-286 and Lines 304-312.

6. In section 3.1.1, line 340, the authors should verify the correct Figure number, “as shown in figure 12” or Figure 5? Similarly, on Line 346, “as shown in figure 13” or Figure 6?

7. In lines 377-378, the sentence lacks a predicate. The correct expression should be: “As shown in Table 2, the foaming data of agents and supercritical CO2 under different water content conditions are presented.” Please ensure grammatical accuracy and clarity when presenting the study.

8. In lines 384-386, the authors mentioned “low water” and “high water”. If the authors mean low water content and high water content, please use more professional terminology to express these concepts.

9.  In section 3.1.4, Lines 412-417, the authors presented visual results of foam effects, but there is a lack of in-depth explanation for these findings. Are these differences due to varying chemical compositions resulting in different interactions in CO2 liquid and supercritical states? Or do they stem from effects on surface tension and/or viscosity? Alternatively, are there other contributing factors? It is necessary for the authors to provide explanations for these results to enhance the completeness and professionalism of the article.

10.  Please ensure consistency in the labeling of figures, such as maintaining uniform capitalization of axis titles in Figure 9. Variations between uppercase and lowercase letters should be avoided to adhere to academic standards for scholarly articles.

11.  For Figures 8, 11, 12, 13, and 14, avoid using “the” in the X-axis and Y-axis titles, as these titles describe general variables.

Comments on the Quality of English Language

The author employs excessively long sentences, which impede readability. The expression lacks logical coherence and contains several notable grammatical errors.

Author Response

Comments 1. In the Introduction, there are some deficiencies in the expression. Specifically, on Lines 50-52, the authors state that “However, for reservoirs with permeability less than 10mD, utilizing the crude oil on the fracture surface by water drive is challenging, and CO2 drive is often used for development.” This statement lacks an explanation of why CO2 drive is more suitable when water drive is challenging in reservoirs with low permeability. For instance, CO2 can dissolve components in crude oil within subsurface reservoirs, thereby reducing the viscosity of the oil and enhancing its mobility, which facilitates oil displacement. Please supplement this statement by providing explanations to make it more complete and logically coherent. The same issue to the statement “For low-permeability reservoirs with less fracture development, CO2 fracturing is often used for development” on Lines 53-54.

Response 1. We agree with this comment. Therefore, we have explained these two points in some detail (lines 50-62).

Comments 2.  I recommend adding a paragraph in the Introduction section to provide a deeper discussion of the significance of this study. Emphasize that this research not only addresses the challenges in developing low-permeability reservoirs but also presents a practical CO2 utilization technology with promising real-world applications. Moreover, highlight how this study aligns with ongoing research in CCUS in China and globally, especially since 2023, thereby underscoring its broader relevance and importance.

Response 2. We agree with this comment. Therefore, we have emphasized. The development limitations of the K<1mD reservoir and the application potential of supercritical CO2, combined with the current common low permeability reservoir development methods, illustrate the necessity of the study in this paper(lines 165-181).

Comments 3. Please review the entire manuscript and correct the subscripts for CO2 and N2.

Response 3. We agree with this comment. Therefore, we have revised all the subscript figures.

Comments 4. In lines 197-200, the authors mentioned the content of mineralized water and listed the ionic species of the water in Table 1. Please provide additional information regarding the source of the water used in this study. Does it come from a specific reservoir? Please include details about the characteristics of this reservoir, including its rock properties, development history, and any distinctive features. This addition will help readers fully understand similar reservoirs where the results of this study might be applicable, facilitating the broader application and dissemination of the study’s related technologies and findings.

Response 4. We agree with this comment. Therefore, we added information such as the permeability and development stage of the reservoir(lines 218-223).

Comments 5.  Some sentences are quite long and complex, significantly affecting readability. For instance, on Lines 277-282, the sentence “in order to avoid the volatilization of oil and water from the side of the core during the heating period, it is necessary to add a perimeter pressure so that the sleeve is stick close to the surface of the core, and at the same time, close the outlet and the inlet of the gripper, to prevent the oil and water vapors that have been evaporated from the end face from escaping out of the gripper” contains 76 words. Breaking it into shorter, more manageable parts can enhance clarity and readability. The authors may consider revising it as follows: “To avoid the volatilization of oil and water from the sides of the core during the heating period, it is necessary to add perimeter pressure. This pressure ensures that the sleeve sticks closely to the surface of the core. Additionally, the outlet and the inlet of the gripper should be closed to prevent the evaporated oil and water vapors from escaping from the end face of the gripper.” Similar improvements can be applied to other lengthy sentences, such as those found on Lines 283-286 and Lines 304-312.

Response 5. We agree with this comment. Therefore, we have broken down the long single sentences in the manuscript(lines 297-304, 311-316, 326-333).

Comments 6. In section 3.1.1, line 340, the authors should verify the correct Figure number, “as shown in figure 12” or Figure 5? Similarly, on Line 346, “as shown in figure 13” or Figure 6?

Response 6. We agree with this comment. Therefore, we have changed Figure 12 in the body to Figure 5 and Figure 3 to Figure 6(lines 361 and 367).

Comments 7. In lines 377-378, the sentence lacks a predicate. The correct expression should be: “As shown in Table 2, the foaming data of agents and supercritical CO2 under different water content conditions are presented.” Please ensure grammatical accuracy and clarity when presenting the study.

Response 7. We agree with this comment. Therefore, we have amended this sentence(lines 398-400).

Comments 8. In lines 384-386, the authors mentioned “low water” and “high water”. If the authors mean low water content and high water content, please use more professional terminology to express these concepts.

Response 8. We agree with this comment. Therefore, we have changed to "high water content" and "low water content".

Comments 9.  In section 3.1.4, Lines 412-417, the authors presented visual results of foam effects, but there is a lack of in-depth explanation for these findings. Are these differences due to varying chemical compositions resulting in different interactions in CO2 liquid and supercritical states? Or do they stem from effects on surface tension and/or viscosity? Alternatively, are there other contributing factors? It is necessary for the authors to provide explanations for these results to enhance the completeness and professionalism of the article.

Response 9. We agree with this comment. However, at present, some specific detection data are lacking, and the differences causing this phenomenon cannot be accurately analyzed for the time being. However, the different foaming phenomena of the agent under liquid CO2 and supercritical CO2 have been explained in lines 372-383.

Comments 10.  Please ensure consistency in the labeling of figures, such as maintaining uniform capitalization of axis titles in Figure 9. Variations between uppercase and lowercase letters should be avoided to adhere to academic standards for scholarly articles.

Response 10. We agree with this comment. Therefore, we have corrected Figure 9.

Comments 11.  For Figures 8, 11, 12, 13, and 14, avoid using “the” in the X-axis and Y-axis titles, as these titles describe general variables.

Response 11. We agree with this comment. Therefore, we have corrected figures 8,11,12,13 and 14.

Reviewer 2 Report

Comments and Suggestions for Authors

The authors investigated the foaming performance of two CO2-soluble polymer-based foaming agents under different water contents, different CO2 phases, and different oil contents. Firstly. It screened out an agent with relatively excellent performance, then carried out the repulsion experiment for the low-permeability fractured cores with the permeability of the cores less than 1 mD to investigate the blocking effect of the agents under different injection modes, which can provide certain theoretical basis. Improvement can be considered from the following aspects:

1. add the foaming principle of this type of blowing agent, which can be illustrated in a drawing;

2. supplement the resistance factor plots at different injection rates, which would be better combined with Table 4;

3. add photographs of the micro-morphology of the foam under oil-containing conditions and to form a comparative illustration in the absence of oil;

4. supplement static foaming experiments with agents with oil content greater than 100:20, such as 100:50, to match the actual conditions in the formation more closely. 

Author Response

Comments 1.add the foaming principle of this type of blowing agent, which can be illustrated in a drawing;

Response 1. We agree with this comment. Therefore, we add it to Figure 3 (b).

Comments 2. supplement the resistance factor plots at different injection rates, which would be better combined with Table 4;

Response 2. We agree with this comment. Therefore, we add it to Figure 11 (b).

Comments 3. add photographs of the micro-morphology of the foam under oil-containing conditions and to form a comparative illustration in the absence of oil;

Response 3. We agree with this comment. However, according to the previous experimental results, the whole system is black after adding crude oil, and the light transmission is inferior, so it is impossible to take clear photos at present.

Comments 4. supplement static foaming experiments with agents with oil content greater than 100:20, such as 100:50, to match the actual conditions in the formation more closely. 

Response 4. We agree with this comment. This test is of great significance to investigate the effect of oil content on foam. However, foam is mainly used in the high water cut stage, and the dominant seepage channels and the water cut are generally higher than 80%; 100:20 is very close to the real formation, which is not considered in this study for the time being, and will be supplemented according to needs in the later in-depth survey.

Round 2

Reviewer 1 Report

Comments and Suggestions for Authors

The manuscript has been improved. It could be accepted in the present form.